# Peritoneal dialysis after failed kidney allograft: Comparing patients with and without pd before transplant

Na Tian[1], Han Meng[1], Winston W. S. Fung[2], Jack K. C. Ng[2], Gordon C. K. Chan[2], Vickie W. K. Kwong[2], Wing-Fai Pang[2], Kai-Ming Chow[2], Philip K. T. Li[2], Cheuk Chun Szeto[2,3]*

1 Department of Nephrology, General Hospital of Ningxia Medical University, Yinchuan, Ningxia, 2 Carol & Richard Yu Peritoneal Dialysis Research Centre, Department of Medicine & Therapeutics, Prince of Wales Hospital, Hong Kong, China, 3 Li Ka Shing Institute of Health Sciences (LiHS), Faculty of Medicine, The Chinese University of Hong Kong, Shatin, Hong Kong, China

* ccszeto@cuhk.edu.hk

## Abstract

**Data Availability Statement:** All relevant data are within the paper and its Supporting Information files.

### Background

The result of published studies on the clinical outcome of peritoneal dialysis (PD) after kidney allograft failure is conflicting. There are also few published data on the outcome of patients who had PD before kidney transplant and then return to PD after allograft failure.

### Methods

We reviewed 100 patients who were started on PD after kidney allograft failure between 2001 and 2020 (failed transplant group); 50 of them received PD before transplant. We compared the clinical outcome to 200 new PD patients matched for age, sex, and diabetic status (control group).

### Results

The patients were followed for 45.8 ± 40.5 months. the 2-year patient survival rate was 83.3% and 87.8% for the failed transplant and control groups, respectively (log rank test, p = 0.2). The corresponding 2-year technique survival rate 66.5% and 71.7% (p = 0.5). The failed transplant and control groups also had similar hospitalization rate and peritonitis rate. In the failed transplant group, there was also no difference in patient survival, technique survival, hospitalization, or peritonitis rate between those with and without PD before transplant. In the failed transplant group, patients who had PD before transplant and then returned to PD after allograft failure had substantial increase in D/P4 (0.585 ± 0.130 to 0.659 ± 0.111, paired t-test, p = 0.032) and MTAC creatinine (7.74 ± 3.68 to 9.73 ± 3.00 ml/min/1.73m$^2$, p = 0.047) from the time before the transplant to the time after PD was resumed after failed allograft.

**Funding:** The work of Dr. Tian Na was supported by the International Association of Chinese Nephrologists scholarship. This study was also supported in part by the Richard Yu Chinese University of Hong Kong (CUHK) PD Research Fund, and CUHK research accounts 6905134 and 7105912. The funders had no role in study design, data collection and analysis, decision to publish, or preparation of the manuscript.

**Competing interests:** The authors have declared that no competing interests exist.

## Conclusions

The clinical outcome of PD patients with a failed kidney allograft is similar to other PD patients. However, patients who have a history of PD before kidney transplant and then return to PD after allograft failure have increased peritoneal transport parameters.

## Introduction

Although kidney transplantation is the ideal treatment of end stage kidney disease (ESKD) [1], 26% to 42% of kidney allografts fail within 10 years after kidney transplantation [2], and long term dialysis is necessary for most of them. Although chronic hemodialysis and peritoneal dialysis (PD) are generally regarded as equivalent in terms of efficacy and clinical outcome [3, 4], there are at least theoretical concerns of using PD for ESKD patients who have kidney allograft failure. For example, the use of immunosuppressive therapy may increase the risk of infection and peritonitis [5–10], and the presence of kidney allograft in the pelvic cavity may limit the cycle volume of PD exchange and dialysis adequacy [11].

There are a number of published studies on the clinical outcome of PD after kidney allograft failure [5, 6, 10, 12–18], but the result is conflicting. For example, the French Language Peritoneal Dialysis Registry (RDPLF) study reported similar patient survival and peritonitis rates but a higher technique failure in the group who started PD after failed kidney transplant [5]. In contrast, another study from Brazil found a 4.4-fold excess in the risk of death in patients who started PD after failed kidney transplant, while the technique survival rate and peritonitis rate were similar to the group without a history of transplantation [6].

To complicate the matter, the outcome of PD *per se* is highly variable in different countries [19, 20]. In Hong Kong, the PD-first policy was adopted since early 1990s [21], and all dialysis centers are proficient in taking care of PD patients, which possibly lead to a favorable outcome [21, 22]. Such policy also provides with us an excellent opportunity to assess an unselected group of patients who started PD after failed kidney transplant, as well as the outcome of a subgroup of patients who had PD before kidney transplant, and then resumed on PD after their kidney allograft failed.

## Patients and methods

### Study design and patient selection

This is a retrospective case-control study approved by the Clinical Research Ethical Committee of the Chinese University of Hong Kong (CRE-2021.367). Written informed consent was obtained from all patients. All study procedures were in compliance with the Declaration of Helsinki. The clinical and research activities being reported are consistent with the Principles of the Declaration of Istanbul as outlined in the 'Declaration of Istanbul on Organ Trafficking and Transplant Tourism'. We reviewed all consecutive patients who had kidney allograft failure and were put on PD in our center from 2001 to 2020 (the failed transplant group). For each of them, we identified two incident PD patients, who were matched in age, sex, and diabetic status, in our center during the same period (the control group).

### Data collection

We reviewed the computerized electronic patient record, followed by manual review of patient medical and nursing notes. Clinical data including demographic information, comorbidity,

and details of the previous transplant were reviewed. Comorbid conditions reviewed included history of diabetes, coronary artery disease, cerebrovascular disease, peripheral vascular disease, chronic pulmonary disease, and previous malignancy. The Charlson's comorbidity index was computed [21, 22].

We also reviewed the result of biochemical tests, as well as peritoneal transport, dialysis adequacy, and nutritional assessment performed 1 to 2 months after the patients were stable on PD. Peritoneal transport characteristic was assessed by the standard peritoneal equilibration test (PET) [23], and was represented by the dialysate-to-plasma ratios of creatinine at 4-hour (D/P4) after correction for glucose interference, as well as the mass transfer area coefficient (MTAC) of creatinine normalized for body surface area (BSA) calculated by a standard formula [24]. BSA was determined by the formula of Gehan and George [25]. For patients in the failed transplant group who were treated with PD before transplant, we also reviewed their peritoneal transport characteristics before kidney transplant, which was performed one to two months after they were first started on PD. Dialysis adequacy was assessed by 24-hour dialysate and urine collection, with the calculation of total Kt/V [26, 27]. Residual glomerular filtration rate (GFR) was calculated as the average of 24-hour urinary urea and creatinine clearance [28]. Nutritional status was represented by serum albumin level, normalized protein nitrogen appearance (NPNA), and fat-free edema-free body mass (FEBM). Serum albumin level was measured by bromcresol purple method. NPNA was calculated by the modified Bergstrom's formula [29]. FEBM was determined by the creatinine kinetic method according to the Forbes and Brunining's formula [30] and adjusted to the percentage of ideal body weight.

## Study end-point

All patients were followed until 31 December 2021. The clinical management was according to the decision of individual nephrologists. For the failed transplant group, immunosuppressive therapy (except glucocorticoid) was generally stopped at the time of PD catheter insertion, while glucocorticoid was gradually tapered off in the subsequent 3 to 6 months. The primary end points of this study were patient survival and technique survival. For patient survival, censoring events include conversion to long-term hemodialysis, kidney transplantation, transfer to other center, or loss to follow up. For technique survival, patient death and transfer to hemodialysis were taken as events, a second transplantation was treated as competing event, and censoring events include transfer to other center or loss to follow up. Secondary end points include number of hospital admission and duration of hospitalization, peritonitis rate, and peritonitis-free survival. Peritonitis was diagnosed according to the criteria recommended by the International Society for Peritoneal Dialysis.

## Statistical analysis

Statistical analysis was performed by SPSS for Windows software version 22.0 (IBM, Armonk, NY). All data were expressed as means ± SD unless otherwise specified. We compared the baseline characteristics and clinical outcome between failed transplant and control groups. Subgroup analysis was further performed within the failed transplant group, comparing patients who did and did not have PD before transplant. Data between groups were compared by the Chi square test, Student's t test, or Mann-Whitney U test as appropriate. The Kaplan-Meier analysis and log rank test were used to compare the patient survival, technique survival, and peritonitis-free survival between groups. Hospitalization and peritonitis rate were compared between groups by Mann-Whitney U test. A p value of less than 0.05 was considered significant. All probabilities were two-tailed.

## Results

We reviewed 100 patients who were started on PD after kidney allograft failure (failed transplant group), and they were compared to 200 incident PD patients matched by age, gender, and diabetic status (the control group). Their baseline clinical and biochemical characteristics are summarized and compared in Tables 1 and 2. In the failed transplant group, 50 patients had PD before they received the kidney transplant. In essence, the failed transplant and control groups were comparable in their baseline demographic and clinical characteristics, but the failed transplant group had higher body mass index, lower proportion of diabetic kidney disease as the underlying renal diagnosis, lower incidence of pre-existing ischemic heart disease, higher hemoglobin, and lower serum albumin level.

In the failed transplant group, the baseline demographic and clinical characteristics were similar between patients who did and did not have PD before transplant, except that patients who had PD before transplant were slightly younger than the others. The clinical characteristics of their kidney transplant are further summarized and compared in Table 3. In essence, kidney allograft failure were more likely caused by rejection, and less likely due to disease recurrence in patients who had PD before transplant.

**Table 1. Demographic and clinical characteristics of the study population.**

|  | control group | failed transplant group | | | P value [1] | P value [2] |
| --- | --- | --- | --- | --- | --- | --- |
|  |  | all case | PD before Tx | no PD before Tx |  |  |
| No. of patient | 200 | 100 | 50 | 50 |  |  |
| male sex, no. of patient (%) | 114 (57%) | 57 (57%) | 24 (48%) | 31 (62%) | p = 1.0 [a] | p = 0.4 [a] |
| age (year) | 51.7 ± 11.7 | 51.8 ± 11.6 | 49.0 ± 12.1 | 54.5 ± 10.5 | p = 1.0 [b] | p = 0.02 [b] |
| body weight (kg) | 62.7 ± 14.5 | 59.7 ± 11.8 | 57.8 ± 11.1 | 61.7 ± 12.3 | p = 0.09 [b] | p = 0.1 [b] |
| body height (cm) | 162.0 ± 8.4 | 162.6 ± 8.5 | 161.8 ± 8.6 | 163.3 ± 8.4 | p = 0.6 [b] | p = 0.4 [b] |
| body mass index (kg/m$^2$) | 20.9 ± 8.9 | 21.4 ± 6.1 | 20.7 ± 6.5 | 22.1 ± 5.7 | p = 0.003 [b] | p = 0.3 [b] |
| primary renal disease, no. of patient (%) |  |  |  |  | p < 0.001 [a] | p = 0.7 [a] |
|  glomerulonephritis | 81 (40.5%) | 68 (68.0%) | 34 (68.0%) | 34 (68.0%) |  |  |
|  diabetic kidney disease | 39 (19.5%) | 4 (4.0%) | 2 (4.0%) | 2 (4.0%) |  |  |
|  hypertensive nephrosclerosis | 23 (11.5%) | 3 (3.0%) | 1 (2.0%) | 2 (4.0%) |  |  |
|  polycystic kidney | 6 (3.0%) | 0 | 0 | 0 |  |  |
|  urological diseases | 10 (5.0%) | 2 (2.0%) | 2 (4.0%) | 0 |  |  |
|  other | 10 (5.0%) | 5 (5.0%) | 3 (6.0%) | 2 (4.0%) |  |  |
|  unknown | 31 (15.5%) | 18 (18.0%) | 8 (16.0%) | 10 (20.0%) |  |  |
| Comorbid conditions, no. of patient (%) |  |  |  |  |  |  |
|  diabetes | 46 (23.0%) | 23 (23.0%) | 11 (22.0%) | 12 (24.0%) | p = 1.0 [a] | p = 1.0 [a] |
|  ischemic heart disease | 27 (13.5%) | 7 (7.0%) | 2 (4.0%) | 5 (10.0%) | p = 0.09 [a] | p = 0.4 [a] |
|  cerebrovascular disease | 29 (14.5%) | 9 (9.0%) | 5 (10.0%) | 4 (8.0%) | p = 0.17 [a] | p = 1.0 [a] |
|  chronic hepatitis B | 26 (13.0%) | 8 (8.0%) | 5 (10.0%) | 3 (6.0%) | p = 0.2 [a] | p = 0.7 [a] |
|  chronic hepatitis C | 4 (2.0%) | 3 (3.0%) | 0 | 3 (6.0%) | p = 0.5 [a] | p = 0.2 [a] |
| Charlson's score | 4.4 ± 2.2 | 4.1 ± 1.9 | 4.0 ± 1.7 | 4.2 ± 2.2 | p = 0.3 [c] | p = 0.7 [c] |

PD, peritoneal dialysis; Tx, transplantation.

[1]between failed transplant group and control groups

[2]between the groups with and without PD before transplant.

Data were compared by

[a]Chi square test

[b]Student's t test, and

[c]Mann-Whitney U test.

**Table 2. Baseline biochemical characteristics of the study population.**

| | control group | failed transplant group | | | P value [1] | P value [2] |
|---|---|---|---|---|---|---|
| | | all case | PD before Tx | no PD before Tx | | |
| no. of patient | 200 | 100 | 50 | 50 | | |
| hemoglobin (g/dl) | 9.4 ± 1.7 | 9.6 ± 1.9 | 9.4 ± 1.7 | 9.8 ± 2.2 | p = 0.04 | p = 0.4 |
| serum albumin (g/L) | 33.4 ± 5.6 | 31.5 ± 6.8 | 31.1 ± 6.8 | 31.9 ± 6.9 | p = 0.01 | p = 0.6 |
| fasting glucose (mmol/L) | 6.8 ± 2.9 | 6.8 ± 2.5 | 6.5 ± 2.1 | 7.1 ± 2.8 | p = 0.2 | p = 0.4 |
| total Kt/V | 2.16 ± 0.75 | 2.13 ± 0.63 | 2.16 ± 0.64 | 2.10 ± 0.63 | p = 0.8 | p = 0.7 |
| residual GFR (ml/min/1.73m$^2$) | 2.70 (1.22–5.45) | 3.60 (0.00–4.90) | 3.18 (0.08–4.69) | 3.67 (0.00–5.68) | p = 0.5* | p = 0.6* |
| NPNA (g/kg/day) | 1.15 ± 0.32 | 1.17 ± 0.32 | 1.16 ± 0.30 | 1.19 ± 0.35 | p = 0.2 | p = 0.7 |
| FEBM (%) | 43.7 ± 12.6 | 46.9 ± 20.2 | 47.6 ± 15.1 | 46.1 ± 24.9 | p = 0.14 | p = 0.7 |

PD, peritoneal dialysis; Tx, transplantation; GFR, residual glomerular filtration rate; NPNA, normalized protein nitrogen appearance; FEBM, fat-free edema-free body mass.

[1]between failed transplant group and control groups

[2]between the groups with and without PD before transplant.

Data were compared by Student's t test or

*Mann-Whitney U test.

**Table 3. Characteristics of the failed transplant group.**

| | all case | PD before Tx | no PD before Tx | P value |
|---|---|---|---|---|
| no. of patient | 100 | 50 | 50 | |
| type of transplant, no. of case (%) | | | | p = 0.8 [a] |
| deceased donor | 86 | 44 (88.0%) | 42 (84.0%) | |
| living donr | 14 | 6 (12.0%) | 8 (16.0%) | |
| duration of previous PD (months) | - | 18 (10–39) | - | - |
| duration of transplant (years) | 12 (5–17) | 11 (2–16) | 14 (9–18) | p = 0.02 [b] |
| immunosuppressive regimens*, no. of case (%) | | | | |
| corticosteroid | 100 | 50 (100.0%) | 50 (100.0%) | p = 1.0 [a] |
| cyclosporine | 62 | 30 (60.0%) | 32 (64.0%) | p = 0.8 [a] |
| tacrolimus | 38 | 20 (40.0%) | 18 (36.0%) | p = 0.8 [a] |
| mycophenolate | 44 | 22 (44.0%) | 22 (44.0%) | p = 1.0 [a] |
| azathioprine | 55 | 28 (56.0%) | 27 (54.0%) | p = 1.0 [a] |
| mTOR inhibitor | 1 | 0 | 1 (2.0%) | p = 1.0 [a] |
| Cause of graft failure, no. of case (%) | | | | p = 0.005 [a] |
| rejection | 38 (38.0%) | 27 (54.0%) | 11 (22.0%) | |
| recurrence | 14 (14.0%) | 4 (8.0%) | 10 (20.0%) | |
| surgical | 1 (1.0%) | 1 (2.0%) | 0 | |
| vascular | 6 (6.0%) | 4 (8.0%) | 2 (4.0%) | |
| multi-factorial | 41 (41.0%) | 14 (28.0%) | 27 (54.0%) | |

PD, peritoneal dialysis; Tx, transplantation; mTOR, mammalian target of rapamycin

Data were compared by

[a]Chi square test and

[b]Mann-Whitney U test.

*before allograft failure

**Table 4. Cause of death of the study population.**

| | control group | failed transplant group | | |
| --- | --- | --- | --- | --- |
| | | all case | PD before Tx | no PD before Tx |
| No. of patient | 200 | 100 | 50 | 50 |
| Death (no. of case) | | | | |
| Total | 69 | 34 | 16 | 18 |
| peritonitis | 9 | 8 | 6 | 2 |
| non-peritonitis infection | 19 | 8 | 1 | 7 |
| ischemic heart disease | 12 | 6 | 4 | 2 |
| sudden cardiac arrest | 10 | 5 | 0 | 5 |
| stroke | 9 | 3 | 1 | 2 |
| termination of dialysis | 4 | 2 | 2 | 0 |
| other specific diseases | 6 | 2 | 2 | 0 |

PD, peritoneal dialysis; Tx, transplantation.

## Patient and technique survival

The patients were followed for 45.8 ± 40.5 months. In the failed transplant group, 34 patients died. The causes of death were summarized and compared to that of the control group in Table 4. During the same period, 24 patients in the failed transplant group were converted to long term hemodialysis, 14 received a second kidney transplant (5 had PD before the first transplant), 4 were transferred to other centers, and 1 lost to follow up. The Kaplan-Meier plots of patient and technique survival are summarized in Fig 1. In essence, the 2-year patient survival rate was 83.3% and 87.8% for the failed transplant and control groups, respectively (log rank test, p = 0.2). The corresponding 2-year technique survival rate 66.5% and 71.7% (p = 0.5). Within the failed transplant group, the 2-year patient survival rates were 85.1% and 81.2% for patients who did and did not have PD before transplant, respectively (p = 0.9), and the corresponding 2-year technique survival rates were 67.9% and 64.8% (p = 0.8). Because of the small number of events and insignificant difference in the univariate analysis, further multi-variable Cox survival analysis was not performed.

## Hospitalization and peritonitis

During the study period, there were 298 hospital admissions for a total of 2263 days in the failed transplant group, and 857 hospital admissions for a total of 5643 days in the control group. The failed transplant and control groups had similar rates of hospital admission (0.83 vs 1.09 per patient-year, p = 0.6) and duration of hospital stay (6.27 vs 7.20 days per patient-year, p = 0.9) (Fig 2). Within the failed transplant group, patients who did and did not have PD before transplant also had similar rates of hospital admission (0.86 vs 0.79 per patient-year, p = 0.9) and duration of hospital stay (7.98 vs 4.46 days per patient-year, p = 0.5) (Fig 2).

During the study period, there were 123 and 238 episodes of peritonitis for the failed transplant and control groups, respectively, and the overall peritonitis rates were 0.34 and 0.30 episodes per patient-year (p = 0.6). Within the failed transplant group, patients who did and did not have PD before transplant had 62 and 61 episodes of peritonitis, respectively, and the corresponding peritonitis rates were 0.33 and 0.35 episodes per patient-year (p = 0.8). The Kaplan-Meier plots of peritonitis-free survival are summarized in Fig 3. In essence, the 2-year peritonitis-free survival rate was 63.2% and 61.1% for the failed transplant and control groups, respectively (p = 0.5). Within the failed transplant group, the 2-year peritonitis-free survival

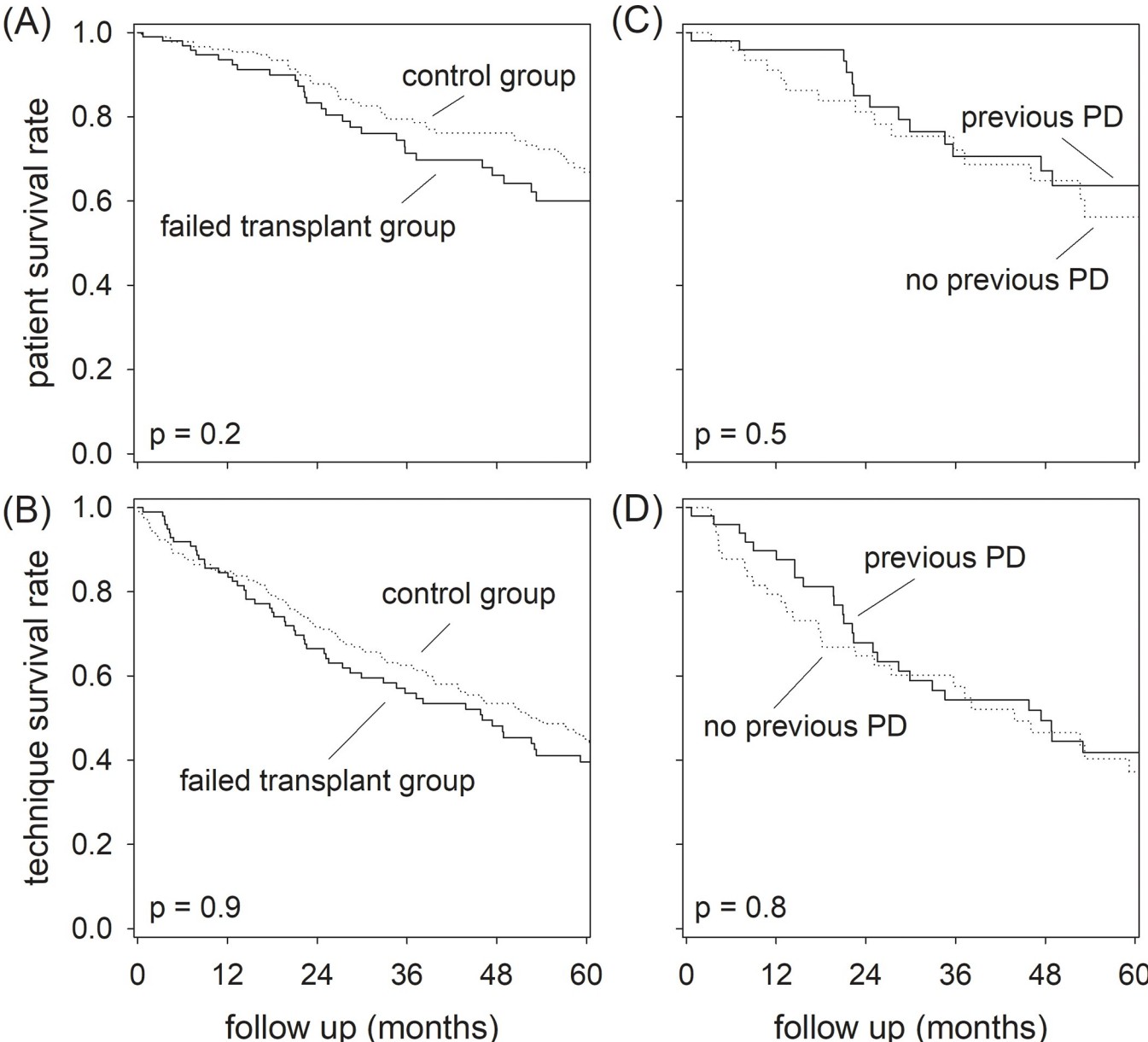

**Fig 1. Patient and technique survival of the study population.** Kaplan Meier plots for (A) patient survival; and (B) technique survival of the failed transplant and control groups, and, within the failed transplant group, Kaplan Meier plot for (C) patient survival; and (D) technique survival for patients who did and did not have peritoneal dialysis (PD) before transplant. For patient survival, censoring events include conversion to long-term hemodialysis, kidney transplantation, transfer to other center, or loss to follow up. For technique survival, patient death and transfer to hemodialysis were taken as events, a second transplantation was treated as competing event, and censoring events include transfer to other center or loss to follow up. Data were compared by the log-rank test.

rates were 65.8% and 60.7% for patients who did and did not have PD before transplant, respectively (p = 0.3).

## Peritoneal transport characteristics

The peritoneal transport characteristics were compared between groups, as summarized in Fig 4. The failed transplant group and control group had similar D/P4 (0.638 ± 0.129 vs

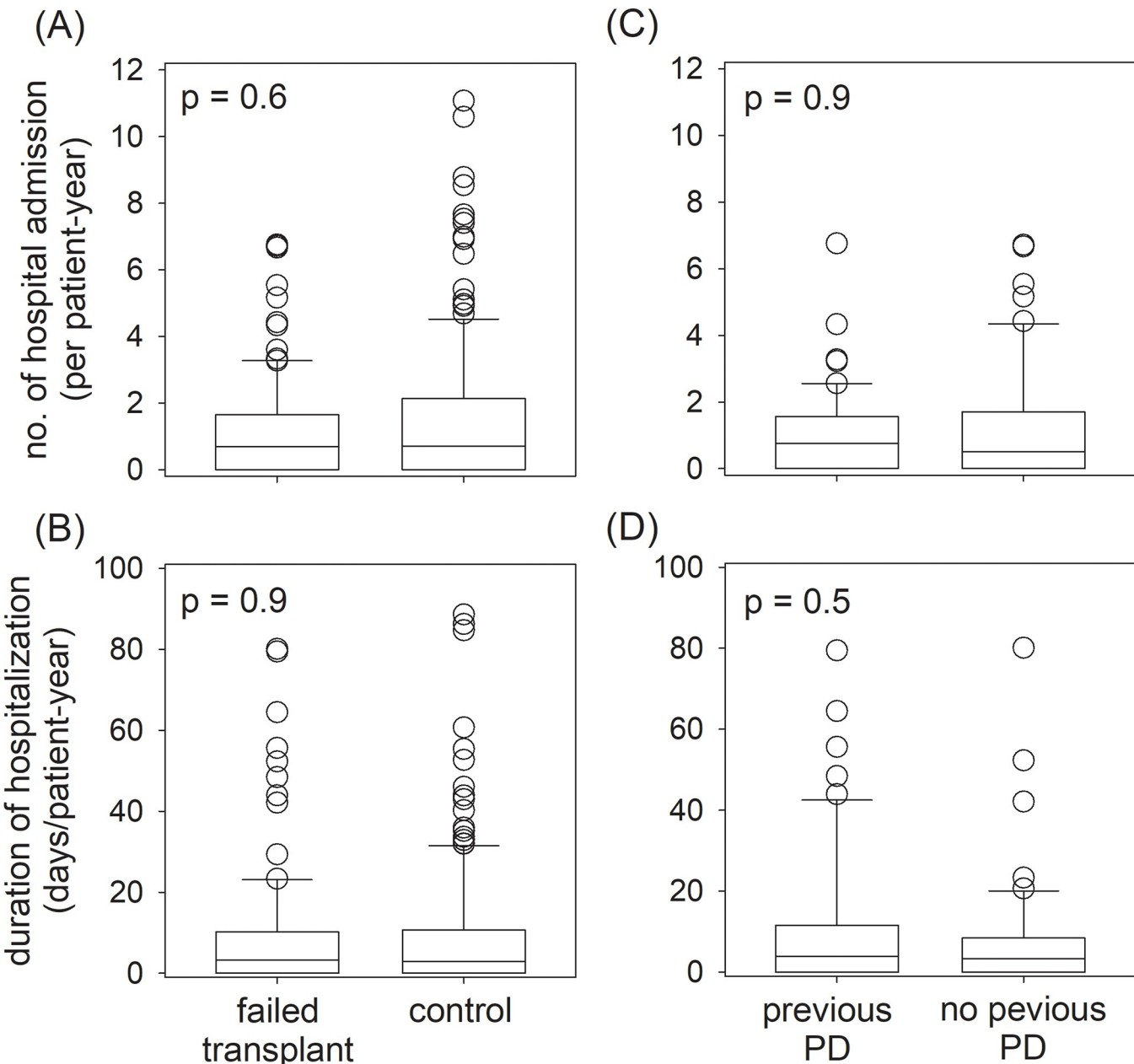

**Fig 2. Hospitalization of the study population.** Comparison between the failed transplant and control groups: (A) number of hospital admission; and (B) duration of hospitalization; and, within the failed transplant group, between patients who did and did not have peritoneal dialysis (PD) before transplant: (C) number of hospital admission; and (D) duration of hospitalization. Data were adjusted to per patient-year of follow up and compared by Mann-Whitney U test.

0.633 ± 0.153, p = 0.8) and MTAC creatinine (9.62 ± 4.57 vs 9.77 ± 5.77 ml/min/1.73m$^2$, p = 0.8). In the failed transplant group, patient who did and did not have PD before transplant also had similar D/P4 (0.654 ± 0.106 vs 0.623 ± 0.146, p = 0.3) and MTAC creatinine (9.88 ± 3.46 vs 9.38 ± 5.44 ml/min/1.73m$^2$, p = 0.6).

For the failed transplant group, however, the subgroup of patients who had PD before transplant and then returned to PD after allograft failure had substantial increase in D/P4 (0.585 ± 0.130 to 0.659 ± 0.111, paired t-test, p = 0.032) and MTAC creatinine (7.74 ± 3.68 to

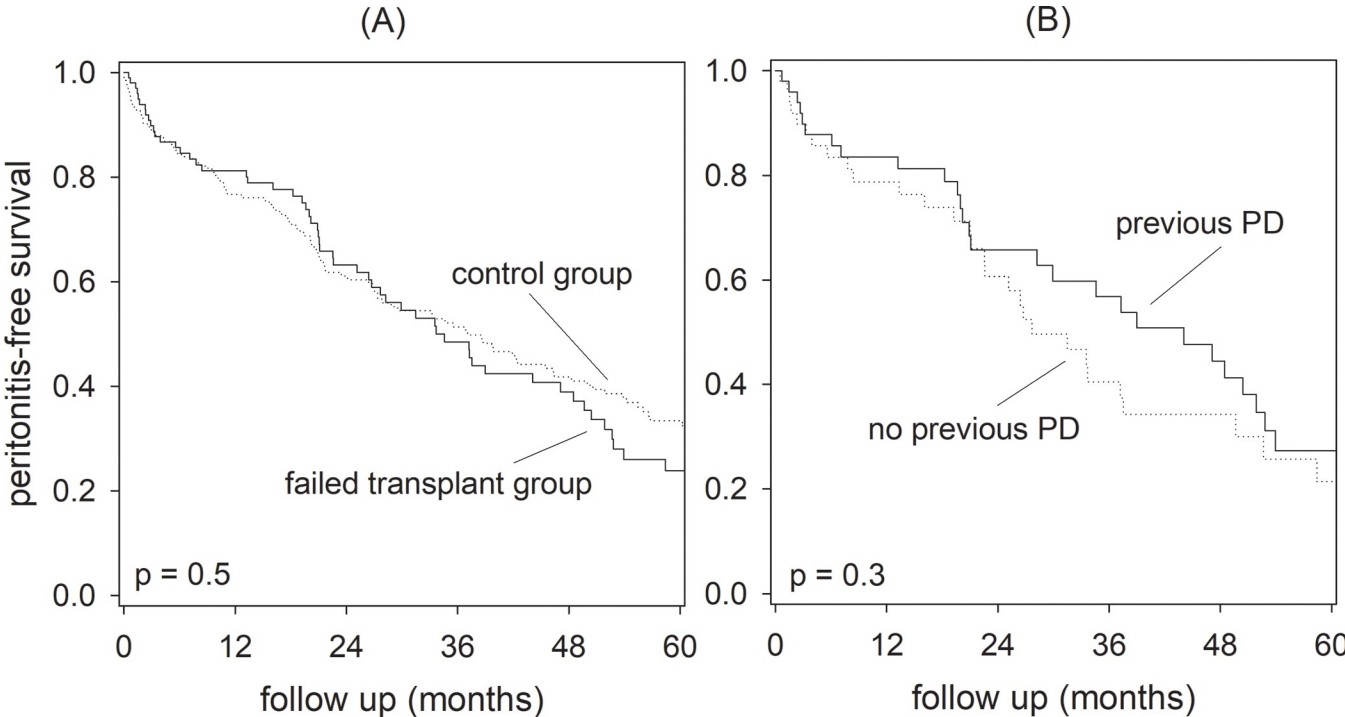

**Fig 3. Peritonitis-free survival of the study population.** Kaplan Meier plots for (A) failed transplant versus control groups; and (B) patients who did and did not have peritoneal dialysis (PD) before transplant. Data were compared by the log-rank test.

9.73 ± 3.00 ml/min/1.73m$^2$, p = 0.047) from the time before the transplant to the time after PD was resumed after failed allograft (Fig 5). Patients who had peritonitis episodes during PD before the kidney transplant had significantly higher increase in D/P4 (0.224 ± 0.134 vs 0.026 ± 0.141, p = 0.006) and MTAC creatinine (6.49 ± 3.42 vs 0.57 ± 4.24 ml/min/1.73m$^2$, p = 0.005) than those who were peritonitis-free. The duration of PD before transplant had a modest inverse correlations with D/P4 (Spearman's r = -0.395, p = 0.011) and MTAC creatinine (r = -0.279, p = 0.08) after allograft failure, although the latter one did not reach statistical significance.

## Discussion

In this study, we found that patient who were started on PD after failed kidney allograft and new PD patients had similar patient and technique survival rates, number of hospital admission, duration of hospital stay, as well as the peritonitis rates. Within the failed allograft group, the clinical outcome was also similar between patients who did and did not have PD before transplant. However, patients who had peritonitis episodes during PD before kidney transplant had significantly higher small solute transport rate than the others.

PD is a valid choices of dialysis modality for the patients following failed kidney allograft, but it is often not considered because of the worry for peritonitis, especially among immunosuppressed patients [5]. In the present analysis, we found similar outcomes between incident PD patients with and without a history of failed kidney allograft. Our result is similar to most but not all other studies, as summarized in Table 5. In essence, all studies except two [6, 12] showed similar patient survival rates between patients with and without a history of kidney transplant. Technique survival rate was reported in 8 studies, of which 6 showed a similar finding. Peritonitis rate was also similar between the groups in all studies except one [13]. In a meta-analysis, Meng et al [31] concluded that new PD patients with a history of failed

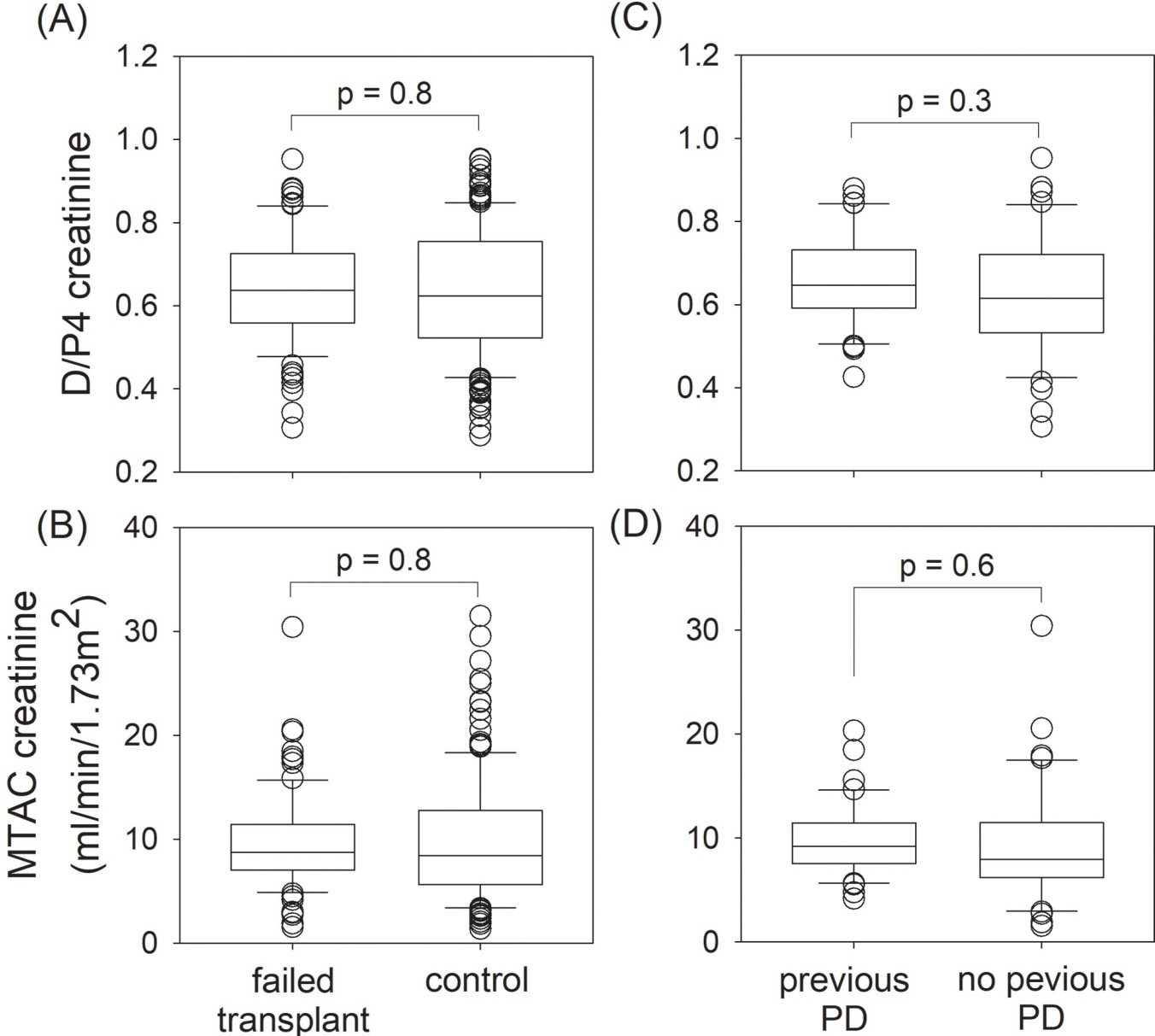

**Fig 4. Peritoneal transport characteristics of the study population.** Comparison between the failed transplant and control groups: (A) dialysate-to-plasma ratios of creatinine at 4-hour (D/P4); and (B) mass transfer area coefficient (MTAC) of creatinine; and, within the failed transplant group, between patients who did and did not have peritoneal dialysis (PD) before transplant: (C) D/P4; and (D) MTAC creatinine. Data were compared by student's t test.

transplant had similar mortality risks as transplant-naïve ones. It is important to note that many previous studies in this area did not match their control group for age and comorbidity load, which may contribute to a better survival in the failed-transplant group [10, 14, 16, 17]. In the present study, we matched the control group by gender, age and diabetes status and found a similar result, which further support the conclusion that a history of failed kidney allograft does not preclude patients from PD.

In the present study, a history of failed transplant was not associated with a higher risk of peritonitis. The overall peritonitis rates were 0.34 and 0.30 episodes per patient-year in the failed transplant and control groups, respectively. An increased incidence of peritonitis among

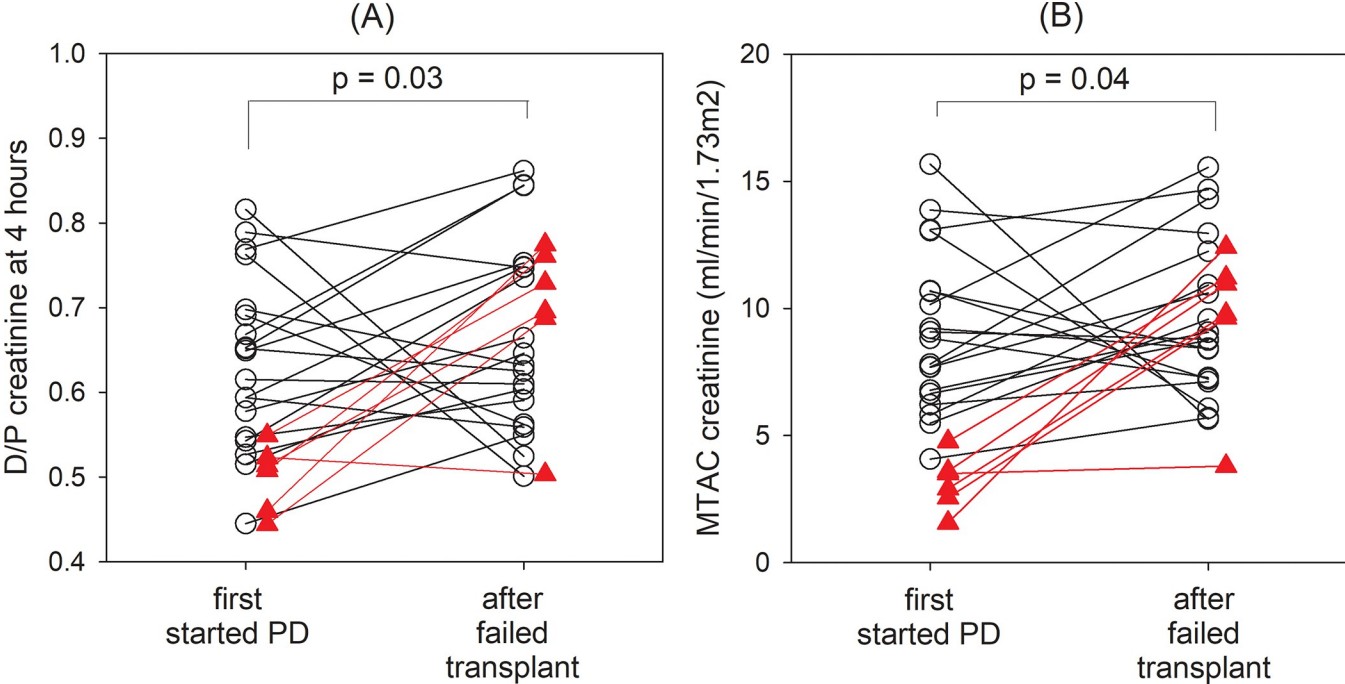

**Fig 5. Change in peritoneal transport characteristics between the time when the patients were newly put on peritoneal dialysis (PD) before kidney transplantation and back on PD after failed transplant.** (A) dialysate-to-plasma ratios of creatinine at 4-hour (D/P4); and (B) mass transfer area coefficient (MTAC) of creatinine. Open circles denote patients without peritonitis episodes before kidney transplantation; closed red triangles denote patients with peritonitis episodes before transplantation. Data were compared by paired Student's t test.

patients with a history of failed transplant was reported in some previous studies [12, 13, 32] but not all [14, 17].

We found no difference in the peritoneal transport characteristics between the failed transplant and control groups, and, within the failed transplant group, between patients who did and did not have PD before transplant. Similar observations were made by Chaudhri et al [18]. In contrast, Wilmer et al. [33] reported that patients with a history of failed transplant were two-times more likely to be high transporters, while Badve et al. [14] found that patients in the failed transplant group had slightly reduced small solute clearance. Animal studies also showed that long-term exposure to calcineurin inhibitor led to peritoneal fibrosis [34]. Taken together, a history of kidney transplant does not appear to have a consistent effect on peritoneal transport characteristics.

A major edge of our present study is we included 50 patients who had PD before kidney transplant, and then returned to PD after their allograft failed. Their baseline characteristics were similar to those who did not have PD before transplant, except that the former group were slightly younger. The two groups also had similar patient survival, technique survival, and hospitalization rates. After failed allograft and returned to PD, their peritoneal transport characteristics were also similar (Fig 4). However, there was actually a small but significant increased small solute transport rate when compared to the time when these patients were first started on PD before transplant (Fig 5), and the change was particularly prominent in patients who had peritonitis episodes before transplant. The absolute change in peritoneal transport parameters, however, were small and did not affect clinical outcome. We believe a history of PD before transplant does not preclude patients from returning to PD after their allograft failed, but careful attention should be paid on possible ultrafiltration problems, especially in patients who had peritonitis episodes before transplant.

**Table 5. Summary of available studies on the outcome of peritoneal dialysis after failed kidney allograft\*.**

| Study | No. of case | No. of control | Follow-up (months) | Patient survival | Technique survival | Peritonitis rate | Hospitalization | Peritoneal transport |
|---|---|---|---|---|---|---|---|---|
| Present study | 100 | 200; by age, gender, and diabetes | 45.8 ± 40.5 | No difference | No difference | No difference | No difference | No difference |
| Benomar et al. [5] | 328 | 656; by age and gender | 17–21 | No difference | Worse | No difference | NR | NR |
| da Costa et al. [6] | 47 | 47; by age, gender, diabetes, PD modality, start year of PD | 14–24 | Worse | No difference | No difference | NR | NR |
| Chen et al. [10] | 445 | 2384; not matched. | NR | NR | NR | No difference | NR | NR |
| Sasal et al. [12] | 42 | 43; by age and diabetes | NR | Worse | NR | No difference | NR | NR |
| Duman et al. [13] | 34 | 82; by age, gender, diabetes, residual renal function, and KT/V | NR | No difference | No difference | Worse | NR | No difference |
| Badve et al. [14] | 309 | 13638; not matched | 12–15 | No difference | No difference | No difference | NR | lower D/P4 |
| Mujais et al. [15] | 494 | 491; by age, gender diabetes, PD modality, cohort year, and dialysis center characteristics | NR | No difference | No difference | No difference | NR | NR |
| Najafi et al. [16] | 43 | 1067; not matched | NR | No difference | No difference | No difference | NR | NR |
| Yang et al. [17] | 47 | 668; not matched. | NR | No difference | No difference | No difference | NR | NR |
| Chaudhri et al. [18] | 50 | 90; by age, gender, diabetes, ethnicity, and year of starting PD | 26 | No difference | Worse | No difference | NR | No difference |

\*As compared to control group (i.e. PD patients without a history of kidney transplant); "worse" indicates that the outcome of the failed allograft group was worse than that of the control group.

PD, peritoneal dialysis; NR, no reported; D/P4, dialysate-to-plasma creatinine ratio at 4 hours.

There remained several limitations in our study. Despite our attempts to match baseline characteristics for the control group, there was inevitably a possibility of unintended selection bias. For example, we matched for the diabetic status while selecting the control group, but substantially more patients in the control group had diabetic kidney disease as the underlying renal diagnosis, indicating that there were more patients in this group had advanced diabetes with multiple end-organ complications. Because of the limitations in our database, we did not analyze the reason for hospitalization. Similarly, we did not have data on the duration required for glucocorticoid replacement therapy, and we were unable to determine the impact of long term steroid replacement on the clinical outcome. Moreover, we also did not review the indication, rate of complication, or the need of transplant graft nephrectomy, or its subsequent impact on patient and technique survival. Further studies are needed to answer these questions.

In conclusion, the clinical outcome of PD patients with a failed kidney allograft is similar to other PD patients. However, patients who have a history of PD before kidney transplant and then return to PD after allograft failure have increased peritoneal transport parameters.

## Supporting information

**S1 Checklist. STROBE statement—Checklist of items that should be included in reports of observational studies.**
(DOCX)

## Author Contributions

**Conceptualization:** Na Tian, Philip K. T. Li, Cheuk Chun Szeto.

**Data curation:** Na Tian, Cheuk Chun Szeto.

**Formal analysis:** Na Tian, Han Meng, Jack K. C. Ng, Gordon C. K. Chan.

**Funding acquisition:** Kai-Ming Chow, Philip K. T. Li, Cheuk Chun Szeto.

**Investigation:** Winston W. S. Fung.

**Methodology:** Winston W. S. Fung, Jack K. C. Ng, Gordon C. K. Chan, Vickie W. K. Kwong.

**Project administration:** Jack K. C. Ng, Cheuk Chun Szeto.

**Resources:** Vickie W. K. Kwong.

**Supervision:** Wing-Fai Pang, Kai-Ming Chow, Philip K. T. Li, Cheuk Chun Szeto.

**Validation:** Winston W. S. Fung, Jack K. C. Ng, Gordon C. K. Chan, Vickie W. K. Kwong, Wing-Fai Pang.

**Writing – original draft:** Na Tian, Han Meng.

**Writing – review & editing:** Cheuk Chun Szeto.

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
