## [Decision Letter · Decision Letter 0]

23 Jan 2023

PONE-D-22-34904Peritoneal Dialysis after Failed Kidney Allograft: Comparing Patients With and Without PD Before TransplantPLOS ONE

Dear Dr. SZETO,

Thank you for submitting your manuscript to PLOS ONE. After careful consideration, we feel that it has merit but does not fully meet PLOS ONE’s publication criteria as it currently stands. Therefore, we invite you to submit a revised version of the manuscript that addresses the points raised during the review process.

ACADEMIC EDITOR: Please correct the manuscript according to Reviewers' comments.

We look forward to receiving your revised manuscript.

Kind regards,

Justyna Gołębiewska

Academic Editor

PLOS ONE

2. Please ensure that you have specified (1) whether consent was informed and (2) what type you obtained (for instance, written or verbal, and if verbal, how it was documented and witnessed). If your study included minors, state whether you obtained consent from parents or guardians. If the need for consent was waived by the ethics committee, please include this information.

Reviewers' comments:

Reviewer's Responses to Questions

**Comments to the Author**

1. Is the manuscript technically sound, and do the data support the conclusions?

Reviewer #1: Yes

Reviewer #2: Partly

2. Has the statistical analysis been performed appropriately and rigorously? 

Reviewer #1: Yes

Reviewer #2: Yes

3. Have the authors made all data underlying the findings in their manuscript fully available?

Reviewer #1: No

Reviewer #2: Yes

4. Is the manuscript presented in an intelligible fashion and written in standard English?

Reviewer #1: Yes

Reviewer #2: Yes

5. Review Comments to the Author

Reviewer #1: Many articles have studied the outcome of PD patients after failed kidney transplantation, but this article is original in that it compares the outcome of these patients to pretransplant treatment. Some patients who have undergone transplantation after PD may want to return to their original PD therapy if transplantation fails : this study suggests that these patients can be safely treated with PD after transplant failure, whether or not they were treated with PD before renal transplantation; however, those who were previously treated with PD before transplantation should have their peritoneal function carefully monitored, as they have higher permeability after transplant failure compared with their pretransplant peritoneal membrane permeability.

The ethical aspects are clearly defined and follow international guidelines.

Article abstract:

The summary is a fair and effective summary of the main text. However, I suggest that the conclusion be the same as the conclusion in the main text. I mean, in the abstract, the authors conclude that all patients have increased patency if they had PD before transplantation, whereas in the conclusion of the article, they are less formal and say that this is favored by a history of PD-related peritonitis. The conclusion of the abstract should be as compelling as the conclusion of the article.

Article :

The article is well structured.

Two minor comments:

- In the last paragraph of the "study end point" section: patient survival is correctly defined, but for technical survival it is written "Technical survival was defined as the patient remaining alive and on PD". There are two different ways of calculating technical survival; it was not clear whether the definition of technique failure included transfer to HD, death or transplantation or whether death and transplantation were censored.

- In the paragraph "Patient and Technique Survival," the authors explain that "in the group of PD patients after failed transplantation, 14 patients received a second kidney transplant." Was the number of secondary renal transplants the same in the subgroup of patients with PD before transplantation compared with those without a history of PD before transplantation, because second transplants may be considered competitive events that could alter the estimate of technical survival ;

The discussion is fine and the limitations of the study are clearly explained.

The conclusion is consistent with the results and discussion, and representative of the fact described in the main text.

References are well selected and generally recent.

Tables and figures are well documented in their legend and support the understanding of the main text. In the legend of figure 1, explain the technique survival (was failure censored on death and transplant or only transplant ?)

Reviewer #2: This study shows that PD therapy is one of the treatment options for patients with ESKD who have a history of failed transplantation, and we believe that it is a very valuable report. However, some points of concern are described below.

Specific comments

1. Please indicate when D/P4 was measured in the first PD in patients with PD before transplant. Is it 1 to 2 months after the patients were stable on PD, similar to the timing of PD after graft failure?

2. Please indicate when Kt/V, residual GFR, NPNA, and FEBM were measured.

3. In Table 4, please indicate in the title that it is a summary of the comparison between transplant and no transplant. Also, please correct the "worse" so that it is clear which is worse, transplant or no transplant.

4. In line 12 of the Discussion 2nd paragraph, you indicate that reference 5 is not a matched study. However, actually reference 5 shows a matched cohort study. Please correct this point.

5. In Tables 1 and 2, for covariates that are clearly not normally distributed such as Charlson's score and duration of PD before transplant, we think the Mann-Whitney test should be used instead of Student's t test. Similarly, when comparing nonparametric data between previous PD and no previous, the Wilcoxon signed-rank test should be used.

6. Please make a list of causes of death and compare both groups if possible.

7. Please validate whether transplantation affects patient and technical survival using a multivariate-adjusted survival analysis in a total cohort combining both groups. Similarly, please validate whether previous PD before transplant affects outcome by adding previous PD as a covariate.

8. Despite being matched by DM status, the ratio of diabetic kidney disease is diverging between the control and all case group. Please describe the reason for this.

Minor comments

1. In Table 1, please indicate the number of men for gender, and indicate % in parentheses.

2. In Table 3, “duration of transplant before PD” should be described as “duration of transplant”. It can be confusing for patients who have had PD before and after transplant.

3. Please describe the unit of duration of hospitalization in Fig 2 in A and C. Is it a month?

4. The reference numbers in Table 4 do not exist in the manuscript. Please correct this point.

6. PLOS authors have the option to publish the peer review history of their article (what does this mean?). If published, this will include your full peer review and any attached files.

Reviewer #1: No

Reviewer #2: **Yes: **Yusuke Kuroki

---

## [Author Response · Author response to Decision Letter 0]

16 Feb 2023

Reviewer 1

The summary is a fair and effective summary of the main text. However, I suggest that the conclusion be the same as the conclusion in the main text. I mean, in the abstract, the authors conclude that all patients have increased patency if they had PD before transplantation, whereas in the conclusion of the article, they are less formal and say that this is favored by a history of PD-related peritonitis. The conclusion of the abstract should be as compelling as the conclusion of the article.

Page 26, last paragraph: As suggested, we revised the wordings of the concluding paragraph and make it in line with the Abstract.

In the last paragraph of the "study end point" section: patient survival is correctly defined, but for technical survival it is written "Technical survival was defined as the patient remaining alive and on PD". There are two different ways of calculating technical survival; it was not clear whether the definition of technique failure included transfer to HD, death or transplantation or whether death and transplantation were censored.

Page 7, paragraph 1, line 3-5: As suggested, we clarify the definition of events for technique survival analysis.

In the paragraph "Patient and Technique Survival," the authors explain that "in the group of PD patients after failed transplantation, 14 patients received a second kidney transplant." Was the number of secondary renal transplants the same in the subgroup of patients with PD before transplantation compared with those without a history of PD before transplantation, because second transplants may be considered competitive events that could alter the estimate of technical survival

Page 7, paragraph 1, line 4-5; page 15, paragraph 1, line 4: We clarify the number of patients who had a second transplant in the subgroup with PD before transplant. We also clarify that second transplantation was treated as a competing event for technique survival analysis.

Tables and figures are well documented in their legend and support the understanding of the main text. In the legend of figure 1, explain the technique survival (was failure censored on death and transplant or only transplant ?)

Figure 1 legend: We elaborate on the definition of events for patient and technique survival as suggested.

Reviewer 2

Please indicate when D/P4 was measured in the first PD in patients with PD before transplant. Is it 1 to 2 months after the patients were stable on PD, similar to the timing of PD after graft failure?

Page 6, paragraph 1, line 8: We clarify that for patients with PD before transplant, the original PET was performed one to two months after they were first started on PD.

Please indicate when Kt/V, residual GFR, NPNA, and FEBM were measured.

Page 5, last line; page 6, line 1: We clarify that Kt/V, residual GFR, NPNA, and FEBM were measured 1 to 2 months after the patients were stable on PD.

In Table 4, please indicate in the title that it is a summary of the comparison between transplant and no transplant. Also, please correct the "worse" so that it is clear which is worse, transplant or no transplant.

Table 5: The title is revised as suggested. We also add a footnote to explain the meaning of “worse”.

In line 12 of the Discussion 2nd paragraph, you indicate that reference 5 is not a matched study. However, actually reference 5 shows a matched cohort study. Please correct this point.

Page 21, last 2 lines: The citation is updated. We are sorry for the mistake.

In Tables 1 and 2, for covariates that are clearly not normally distributed such as Charlson's score and duration of PD before transplant, we think the Mann-Whitney test should be used instead of Student's t test. Similarly, when comparing nonparametric data between previous PD and no previous, the Wilcoxon signed-rank test should be used.

Table 1, 2, and 3: We revised the analysis for Charlson’s score, and residual GFR by non-parametric test as suggested. The data between groups with and without previous PD were also compared by non-parametric test.

Please make a list of causes of death and compare both groups if possible.

Table 4: The cause of death is summarized in a new table as recommended.

Please validate whether transplantation affects patient and technical survival using a multivariate-adjusted survival analysis in a total cohort combining both groups. Similarly, please validate whether previous PD before transplant affects outcome by adding previous PD as a covariate.

Page 15, last 3 lines: Because of the small number of events and insignificant difference in the univariate analysis, further multi-variable Cox survival analysis was not performed.

Despite being matched by DM status, the ratio of diabetic kidney disease is diverging between the control and all case group. Please describe the reason for this.

Page 26, paragraph 1, line 3-6: We clarify that we matched the case and control groups by diabetic status, but it turns out that more patients in the control group had diabetic kidney disease as the underlying cause of kidney failure. The potential selection bias is elaborated in the discussion.

In Table 1, please indicate the number of men for gender, and indicate % in parentheses.

Table 1: We describe the number of men for gender as suggested.

In Table 3, “duration of transplant before PD” should be described as “duration of transplant”. It can be confusing for patients who have had PD before and after transplant.

Table 3: The phrase is revised to “duration of transplant” as suggested.

Please describe the unit of duration of hospitalization in Fig 2 in A and C. Is it a month?

Figure 2: We updated the unit of duration of hospitalization in the figure axis.

The reference numbers in Table 4 do not exist in the manuscript. Please correct this point.

Table 5: We updated the reference number in the table. We are sorry for the mistake in the previous manuscript.

---

## [Decision Letter · Decision Letter 1]

6 Mar 2023

PONE-D-22-34904R1

Peritoneal Dialysis after Failed Kidney Allograft: Comparing Patients With and Without PD Before Transplant

PLOS ONE

Dear Dr. SZETO,

Thank you for submitting your manuscript to PLOS ONE. After careful consideration, we feel that it has merit but does not fully meet PLOS ONE’s publication criteria as it currently stands. Therefore, we invite you to submit a revised version of the manuscript that addresses the points raised during the review process.

Please correct according to Reviewer's comments.

We look forward to receiving your revised manuscript.

Kind regards,

Justyna Gołębiewska

Academic Editor

PLOS ONE

Journal Requirements:

Reviewers' comments:

Reviewer's Responses to Questions

**Comments to the Author**

1. If the authors have adequately addressed your comments raised in a previous round of review and you feel that this manuscript is now acceptable for publication, you may indicate that here to bypass the “Comments to the Author” section, enter your conflict of interest statement in the “Confidential to Editor” section, and submit your "Accept" recommendation.

Reviewer #1: All comments have been addressed

Reviewer #2: (No Response)

2. Is the manuscript technically sound, and do the data support the conclusions?

Reviewer #1: Yes

Reviewer #2: Yes

3. Has the statistical analysis been performed appropriately and rigorously? 

Reviewer #1: Yes

Reviewer #2: Yes

4. Have the authors made all data underlying the findings in their manuscript fully available?

Reviewer #1: No

Reviewer #2: Yes

5. Is the manuscript presented in an intelligible fashion and written in standard English?

Reviewer #1: Yes

Reviewer #2: Yes

6. Review Comments to the Author

Reviewer #1: (No Response)

Reviewer #2: The authors adequately answered to the raised points. However, the reviewer will note the following point.

1. In Table 4, the total of the number of PD and no PD before transplantation does not accord with the number of all cases.

2. In Fig 2 B and D, please change the unit to “days/patient-year”.

7. PLOS authors have the option to publish the peer review history of their article (what does this mean?). If published, this will include your full peer review and any attached files.

Reviewer #1: No

Reviewer #2: **Yes: **Yusuke Kuroki

---

## [Author Response · Author response to Decision Letter 1]

14 Mar 2023

14 March 2023

Editor-in-Chief

PLoS One

Sir,

Re: ‘Peritoneal Dialysis after Failed Kidney Allograft: Comparing Patients With and Without PD Before Transplant’ (PONE-D-22-34904R1)

Thanks for your letter on 7th March. The following revisions have been made according to your recommendations:

Reviewer 2

In Table 4, the total of the number of PD and no PD before transplantation does not accord with the number of all cases.

Table 4: The numbers are checked and one typo rectified. We are sorry for the mistake.

In Fig 2 B and D, please change the unit to “days/patient-year”.

Figure 2B and 2D: The label of the y-axis is changed as suggested.

In addition, the information of reference #21 is updated.

We would like to resubmit the manuscript, with the changes in red, for consideration of publication in the PLoS One. Thank you for reviewing our article and we look forward to hearing your favorable reply.

Yours faithfully,

CC Szeto

For N Tian, H Meng, WWS Fung, JKC Ng, GCK Chan, VWK Kwong, WF Pang, KM Chow, and PKT Li

---

## [Decision Letter · Decision Letter 2]

27 Mar 2023

Peritoneal Dialysis after Failed Kidney Allograft: Comparing Patients With and Without PD Before Transplant

PONE-D-22-34904R2

Dear Dr. SZETO,

We’re pleased to inform you that your manuscript has been judged scientifically suitable for publication and will be formally accepted for publication once it meets all outstanding technical requirements.

Kind regards,

Justyna Gołębiewska

Academic Editor

PLOS ONE

Additional Editor Comments (optional):

Reviewers' comments:

Reviewer's Responses to Questions

**Comments to the Author**

1. If the authors have adequately addressed your comments raised in a previous round of review and you feel that this manuscript is now acceptable for publication, you may indicate that here to bypass the “Comments to the Author” section, enter your conflict of interest statement in the “Confidential to Editor” section, and submit your "Accept" recommendation.

Reviewer #2: All comments have been addressed

2. Is the manuscript technically sound, and do the data support the conclusions?

Reviewer #2: Yes

3. Has the statistical analysis been performed appropriately and rigorously? 

Reviewer #2: Yes

4. Have the authors made all data underlying the findings in their manuscript fully available?

Reviewer #2: Yes

5. Is the manuscript presented in an intelligible fashion and written in standard English?

Reviewer #2: Yes

6. Review Comments to the Author

Reviewer #2: (No Response)

7. PLOS authors have the option to publish the peer review history of their article (what does this mean?). If published, this will include your full peer review and any attached files.

Reviewer #2: **Yes: **Yusuke Kuroki

---

## [Editor Report · Acceptance letter]

30 Mar 2023

PONE-D-22-34904R2 

Peritoneal dialysis after failed kidney allograft: comparing patients with and without pd before transplant 

Dear Dr. Szeto:

I'm pleased to inform you that your manuscript has been deemed suitable for publication in PLOS ONE. Congratulations! Your manuscript is now with our production department. 

Kind regards, 

on behalf of

Dr. Justyna Gołębiewska 

Academic Editor

PLOS ONE